# Protective Effect of *Alpinia oxyphylla* Fruit against *tert*-Butyl Hydroperoxide-Induced Toxicity in HepG2 Cells via Nrf2 Activation and Free Radical Scavenging and Its Active Molecules

**DOI:** 10.3390/antiox11051032

**Published:** 2022-05-23

**Authors:** Chae Lee Park, Ji Hoon Kim, Je-Seung Jeon, Ju-hee Lee, Kaixuan Zhang, Shuo Guo, Do-hyun Lee, Eun Mei Gao, Rak Ho Son, Young-Mi Kim, Gyu Hwan Park, Chul Young Kim

**Affiliations:** 1College of Pharmacy and Institute of Pharmaceutical Science and Technology, Hanyang University, Ansan 15588, Korea; cherry890709@huons.com (C.L.P.); gg890718@gmail.com (J.H.K.); jsjeoncy@gmail.com (J.-S.J.); 0702leeeun@naver.com (J.-h.L.); kaixuanzhanggg@163.com (K.Z.); guoshou8080@hanyang.ac.kr (S.G.); do247@naver.com (D.-h.L.); ennmei203@163.com (E.M.G.); sonnaco@huons.com (R.H.S.); ymikim12@hanyang.ac.kr (Y.-M.K.); 2R&D Center, Huons Co., Ltd., Ansan 15588, Korea; 3College of Pharmacy, Kyungpook National University, Daegu 41566, Korea

**Keywords:** *Alpinia oxyphylla*, antioxidant activity, eudesma-3,11-dien-2-one, yakuchinone A

## Abstract

*Alpinia oxyphylla* Miq. (Zingiberaceae) extract exerts protective activity against *tert*-butyl hydroperoxide-induced toxicity in HepG2 cells, and the antioxidant response element (ARE) luciferase activity increased 6-fold at 30 μg/mL in HepG2 cells transiently transfected with ARE-luciferase. To identify active molecules, activity-guided isolation of the crude extract led to four sesquiterpenes (**1**, **2**, **5**, **6**) and two diarylheptanoids (**3** and **4**) from an *n*-hexane extract and six sesquiterpenes (**7**–**12**) from an ethyl acetate extract. Chemical structures were elucidated by one-dimensional, two-dimensional nuclear magnetic resonance (1D-, 2D-NMR), and mass (MS) spectral data. Among the isolated compounds, eudesma-3,11-dien-2-one (**2**) promoted the nuclear accumulation of nuclear factor (erythroid-derived 2)-like 2 (Nrf2) and increased the promoter property of the ARE. Diarylheptanoids, yakuchinone A (**3**), and 5′-hydroxyl-yakuchinone A (**4**) showed radical scavenging activity in 2,2-diphenyl-1-picrylhydrazyl (DPPH) and 3-ethylbenzothiazoline-6-sulphonic acid (ABTS) assays. Furthermore, optimization of extraction solvents (ratios of water and ethanol) was performed by comparison of contents of active compounds, ARE-inducing activity, radical scavenging activity, and HepG2 cell protective activity. As a result, 75% ethanol was the best solvent for the extraction of *A. oxyphylla* fruit. This study demonstrated that *A. oxyphylla* exerted antioxidant effects via the Nrf2/HO-1 (heme oxygenase-1) pathway and radical scavenging along with active markers eudesma-3,11-dien-2-one (**2**) and yakuchinone A (**3**).

## 1. Introduction

Excessive production of reactive oxygen species (ROS) may cause the degeneration of DNA, protein, and lipids that may be a contributing factor in the development of chronic diseases, including cancer, diabetes, atherosclerosis, and arthritis [1,2]. Strategies for protecting against oxidative damage are (i) scavenging ROS using antioxidants including polyphenols, carotenoids, and vitamins, (ii) up-regulation of endogenous antioxidant/phase II detoxifying enzymes through the induction of antioxidant signaling pathways such as the NF-E2-related factor-2 (Nrf2)/antioxidant response element (ARE) signaling pathway [3].

*tert*-Butylhydroperoxide (*t*-BHP) can be metabolized to free radical intermediates by cytochrome P 450 in hepatocytes [4]. Free radical intermediates generated by *t*-BHP can subsequently lead to oxidative-induced hepatocyte damage [5]. Therefore, *t*-BHP has often been used as an oxidative stress injury with in vitro models to identify antioxidant molecules from natural products [6,7].

*Alpinia oxyphylla* Miq. (Zingiberaceae) is widely cultivated and distributed across subtropical regions. The fruit of *A. oxyphylla* has been used as a traditional medicine in Korea, China, and Japan for the treatment of gastrointestinal disorders, urosis, diuresis, ulceration, and dementia [8,9]. Previous pharmacological investigations have indicated that *A. oxyphylla* fruit possesses various attributes, including anti-inflammatory [10] anti-allergy [11], anti-ulcer [12], and neuroprotective activities [13]. Various classes of chemicals are reported in *A. oxyphylla* fruit, including diarylheptanoids, sesquiterpenes, diterpenes, flavonoids, and steroids [9,14].

In this study, bioassay-guided isolation for ARE-inducing constituents from *A. oxyphylla* fruit led to four sesquiterpenes (**1**, **2**, **5**, **6**) and two diarylheptanoids (**3**, **4**) from *n*-hexane extract and six sesquiterpenes (**7**–**12**) from ethyl acetate extract. Among them, eudesma-3,11-dien-2-one (**2**) promoted the nuclear accumulation of Nrf2 and increased the promoter property of the antioxidant response element, and yakuchinone A (**3**) and 5′-hydroxyl-yakuchinone A (**4**) showed radical scavenging activity in DPPH and ABTS assay. In addition to these results, the optimization of solvent extracts for active constituents, radical scavenging activity, and preventive effects on *t*-BHP-induced HepG2 cells damage were also investigated. As a result, 75% ethanol was the best solvent composition for the extraction of *A. oxyphylla* fruit.

## 2. Materials and Methods

### 2.1. Plant Material

*A. oxyphylla* fruit was purchased from a Kyungdong Oriental Herb Market, Seoul, Korea, in June 2017 and identified by one of the authors (C.Y. Kim). A voucher specimen was deposited in the Herbarium of the College of Pharmacy, Hanyang University (HYU-AO-2017-06-01).

### 2.2. Extraction and Isolation of Compounds

*A. oxyphylla* fruit (1.2 kg) were ground into powder and extracted three times with methanol (3 L) for 90 min using an ultrasonic apparatus and subsequently filtered *in vacuo* through a defatted cotton filter. The filtered solute was then concentrated to dryness (126 g) by rotary evaporation under reduced pressure at 40 °C. Next, a crude methanol extract was suspended in 1000 mL water and successively partitioned with *n*-hexane, ethyl acetate (EtOAc), and *n*-butanol (1000 mL × 3 times) to obtain 36.5 g of *n*-hexane extract, 33.9 g of EtOAc extract, 12.5 g of *n*-butanol and 43.1 g of water extract, which was then stored in a freezer (−20 °C). The *n*-hexane extract was isolated by centrifugal partition chromatography (CPC) with a two-phase solvent system of *n*-hexane-EtOAc-*n*-butanol-water (7:3:7:3, *v*/*v*) (Appendix A). Firstly, the rotor was entirely filled, with the lower phase as the stationary phase. Subsequently, the upper phase was pumped into the inlet of the rotor at a flow rate of 10 mL/min while the apparatus ran at 1000 rpm. After reaching hydrostatic equilibrium, samples (10 g) were subjected to the CPC apparatus. The effluents were monitored at 254 nm, and nine fractions (Frs. A–I) were obtained according to the chromatographic profile. The CPC method was operated three times, and the same fractions were combined. For further purification of the Frs. C, D, H, and I, preparative high-performance liquid chromatography (prep-HPLC) was performed with an Inno C18 column (20 × 250 mm, 5 μm, YoungJin Biochrom, Seongnam, Korea) under isocratic conditions with 60% acetonitrile. After prep-HPLC, compounds **1**–**6** were obtained with high purities: compound **1** (181.6 mg) from Fr. C; compound **2** (6.2 mg) from Fr. D; compounds **3** (488.5 mg) and **4** (7.1 mg) from Fr. H. Compounds **5** (121.6 mg) and **6** (40.7 mg) were isolated from Fr. H after applying a silica gel column with *n*-hexane-ethyl acetate (7:1) and subsequent prep-HPLC under isocratic 60% acetonitrile elution. A detailed isolation scheme is described in Appendix A.

The EtOAc extract (32.8 g) was fractionated using Diaion HP-20 (Mitsubishi Chemical Co., Tokyo, Japan) resin with stepwise gradients of 10–100% methanol and monitored by analytical HPLC. The same compositions were combined (60%, 70%, and 80% methanol eluent) and purified on silica gel using a gradient elution of *n*-hexane-ethyl acetate (2:1 → 1:1). After silica gel column chromatography, further isolation was performed by prep-HPLC to result with six compounds: **7** (12.2 mg), **8** (12.1 mg), **9** (202 mg), **10** (8.2 mg), **11** (13.3 mg), and **12** (8.5 mg). A detailed isolation scheme is described in Appendix A, and detailed spectral data of isolated compounds **1**–**12** were depicted in Appendix A.

### 2.3. Cell Culture

HepG2 cells were purchased from ATCC (Manassas, VA, USA) and cultured in Dulbecco’s modified Eagle’s medium supplemented with 10% fetal bovine serum, 100 U/mL penicillin, and 100 μg/mL streptomycin in a humidified atmosphere of 5% CO_2_ at 37 °C.

### 2.4. Cell Viability Assay

One day after plating in 96-well plates (3 × 10^4^ cells), the cells were incubated with indicated concentrations for 24 h after overnight serum starvation and subsequent exposure to 100 µL 3-(4,5-dimethylthiazol-2-yl)-2,5-diphenyl tetrazolium bromide (MTT, Sigma Chemical Co., St. Louse, MO, USA) solution (0.5 mg/mL) for 30 min at 37 °C. The culture medium was removed, and intracellular formazan crystals were dissolved in 300 µL of dimethyl sulfoxide (DMSO). The absorbance was measured at 570 nm with an Infinite M200 PRO microplate reader (Tecan, Salzburg, Austria). All experiments were performed in triplicate.
Cell viability (%) = [(sample-blank)/(control-blank)] × 100

### 2.5. ARE-Luciferase Assay

HepG2 cells stably transfected with an ARE-driven reporter gene construct, pGL4.37[luc2P/ARE/Hygro], were kindly donated by Dr. IJ Cho (Daegu Haany University, Gyeongsan, Korea), who established the stable transfection of HepG2 cells with pGL4.37[luc2P/ARE/Hygro] construct as previously described [15]. HepG2 cells stably transfected with pGL4.37 were plated at a density of 1 × 10^5^ cells/well in 24-well plates. After 24 h, cells were serum-starved overnight and subsequently treated with vehicle, extracts, and compounds for 12 h. Crude extract and compounds were dissolved in DMSO and added to the medium. The final concentration of DMSO did not exceed 0.1%, which did not affect cell cytotoxicity. Sulforaphane (5 μM) (Calbiochem, Darmstadt, Germany) was used as a positive control. After treatment, cells were washed twice with ice-cold phosphate-buffered saline and lysed in a passive lysis buffer (Promega, Madison, WI, USA). The levels of ARE-luciferase activity in the resultant cell lysates were measured using the luciferase assay system (Promega, Madison, WI, USA) according to the manufacturer’s instructions.

### 2.6. Cytoprotective Effects of t-BHP-Induced HepG2 Cells

HepG2 cells were seeded in 96-well plates at a density of 7 × 10^4^ cells/well. After 24 h, the culture medium (200 μL/well) was replaced with a serum-free medium (200 μL/well) containing various concentrations of extract (3, 10, and 30 μg/mL) or compounds **2**–**4** (3, 10, and 30 μM). After 12 h, the cells were treated with 300 μM *t*-BHP to induce oxidative stress for 3 h. The protective effect of extract or compounds was evaluated using an MTT assay. All experiments were performed at least in triplicate.

### 2.7. Preparation of Cell Lysates and Western Blotting

HepG2 cells were plated in 6-well plates at a density of 5 × 10^5^ cells/well. After 24 h, the cells were incubated with different concentrations of extract or compounds for the indicated time after overnight serum starvation. To obtain the nuclear and cytosolic cell lysates and detect heme oxygenase-1 (HO-1) and Nrf2, cells were incubated with 80 μL cold lysis buffer A, containing 10 mM-HEPES-KOH (pH 7.9), 10 mM-KCl, 1.5 mM-MgCl_2_, 1 mM-dithiothreitol, 0.1% nonidet P-40, and inhibitor stock III (1:200) for 10 min on ice. Next, samples were centrifuged at 7200× *g*, 4 °C for 5 min, and the supernatant fractions (cytosolic cell extracts) were collected to determine protein concentration. Pellets were re-suspended in 40 μL cold lysis buffer B, composed of 10 mM HEPES-KOH (pH 7.9), 25% glycerol, 400 mM-KCl, 0.1 mM-EDTA, and inhibitor III (1:200), and then kept on ice for 60 min. Samples were centrifuged at 7200× *g*, 4 °C for 10 min, and the supernatant fractions (nuclear cell extracts) were collected. Both cell extracts were analyzed for protein concentration and stored at −80 °C until use for Western blotting. Equal amounts (20 μg) were separated by 7.5% sodium dodecyl sulfate–polyacrylamide gel electrophoresis and transferred onto nitrocellulose membranes (Millipore, Bedford, MA, USA). The membranes were washed with Tris-buffered saline (10 mM Tris–HCl, 150 mM NaCl, pH 7.5), followed by blocking with 5% (*w*/*v*) non-fat dried milk. The membranes were incubated overnight with antibodies against HO-1, Nrf2, Lamin A/C, and actin at 4 °C. The membranes were washed three times with TBST buffer and then exposed to secondary antibodies coupled to horseradish peroxidase for 2 h at room temperature. ECL chemiluminescence reagents (Pierce; Thermo Fisher Scientific, Inc., Waltham, MA, USA) were used to detect the protein bands. Densitometric analysis of the data obtained from at least three independent experiments was performed using Fusion Solo chemiluminescence and fluorescence imaging system (Vilber-Lourmat, Eberhardzell, Germany).

### 2.8. Extraction for Optimization of Ethanol/Water Solvent Ratios

Fifty grams of *A. oxyphylla* powder were extracted with 100 mL of the following solvents: 100% water, 25% aqueous ethanol, 50% aqueous ethanol, 75% aqueous ethanol, and 100% ethanol for 2 h under reflux. The extract was filtered and dried under a rotary evaporator and freeze-dried.

### 2.9. HPLC Analysis

Each extract was dissolved at a concentration of 10 mg/mL in methanol. Isolated compounds **1**–**12** were used to prepare standard solutions. Each compound was dissolved at 1000 μg/mL concentration in methanol, and serially diluted (1000, 500, 250, 125, 62.5, 6.25, 3.125, 1.5625 and 0.78125 μg/mL). Linear regression equations were calculated with y = ax ± b, where x was the concentration and y was the peak area of each compound. Linearity was established by the coefficient of equation (R^2^). HPLC analysis was performed using an Agilent Technologies 1200 system equipped with an automatic injector, a column oven, and a DAD detector. An Inno C18 column (250 × 4.6 mm, 5 μm, YoungJin Biochrom, Seongnam, Korea) was used. The temperature was maintained at 40 °C, with an injection volume of 20 μL at a flow rate of 1 mL/min. The mobile phase was composed of acetonitrile containing 0.1% formic acid (A) and water containing 0.1% formic acid (B). The gradient elution conditions were as follows: initial 0 min A:B (15:85, *v*/*v*), 12 min A:B (23:77), 25 min A:B (23:77), 35 min A:B (30:70), and 85 min A:B (100:0). The detection wavelength was 230 nm or 254 nm.

### 2.10. DPPH Radical Scavenging Activity

The radical scavenging activity of 2,2-diphenyl-1-picrylhydrazyl (DPPH) was performed according to the method from Shim and Lee [16] that was slightly modified. In brief, 5 μL samples (final concentration: 25, 12.5, 6.25, 3.125 μg/mL in ethanol) were mixed with 100 μL of 400 mM DPPH in 100% (*v*/*v*) ethanol and 95 μL ethanol. After 1 h incubation in the dark covered with aluminum foil, a decrease in absorbance was monitored at 515 nm. The control consisted of 5 μL ethanol. Ethanol was used as a solvent, and trolox (final concentration: 1000, 500, 250, 125, 62.5 μg/mL) was used as a positive control. All samples were analyzed in triplicate.

### 2.11. ABTS Radical Scavenging Activity

The radical scavenging activity of 2,2-azino-bis(3-ethylbenzothiazoline-6-sulfonic acid) diammonium salt (ABTS) was assessed using a modified method [17]. In brief, ABTS (7 mM) and potassium persulfate (2.45 mM) were mixed and heated to 68 °C for 30 min with aluminum foil. After heating, the stock solution was diluted to 1/50 using distilled water. The reactants, 195 μL of the diluted ABTS radical solution and 5 μL of the sample (final concentration: 25, 12.5, 6.25, 3.125 μg/mL in ethanol), were mixed in 96-well plates. Absorbance was immediately read at 723 nm using a microplate reader. Ethanol was used as a solvent, and trolox (final concentration: 1000, 500, 250, 125, and 62.5 μg/mL) was used as a positive control. All samples were analyzed in triplicate.

### 2.12. Statistical Analysis

All data are reported as means ± S.D. The statistical significance of differences between treatments was assessed using Student’s *t*-test. Probability values less than * < 0.05, ** < 0.01 were considered significant.

## 3. Results and Discussion

### 3.1. Bioactivity-Guided Isolation of Compounds

Treatment with 3, 10, and 30 μg/mL of methanolic extract enhanced ARE-luciferase activities with dose-dependent manners. Relative ARE-luciferase activities by 3, 10, and 30 μg/mL of crude extract were enhanced 1.95 ± 0.14-, 3.35 ± 0.24-, and 5.83 ± 0.22-fold, respectively. When 5 μM of sulforaphane was treated as a positive control, ARE-luciferase activity was augmented 11.04 ± 0.42-fold (Figure 1B). Therefore, bioassay-guided fractionation was carried out using an ARE-driven luciferase assay in HepG2 cells to purify the ARE-inducing constituents from *A. oxyphylla* fruits. The crude methanol extract was successively partitioned with *n*-hexane, ethyl acetate, and *n*-butanol. Statistical significances of differences were observed in 3, 10, and 30 μg/mL of *n*-hexane extract and EtOAc extract-treated cells as compared with control cells (Figure 1C). Relative luciferase activities by treatment with 3, 10, and 30 μg/mL of *n*-hexane extract were 2.05 ± 0.29-, 6.18 ± 0.76-, and 11.60 ± 1.35-fold of control cells, respectively. The treatment of ethyl acetate extract with 3, 10, and 30 μg/mL also induced ARE-luciferase activities of 1.97 ± 0.18-, 4.10 ± 0.56-, and 12.67 ± 2.23-fold. In experiments conducted together, sulforaphane (5 μM) enhanced 26.29 ± 4.62-fold (Figure 1C).

HPLC analysis revealed that the crude extract contained several major compounds (Figure 1A). The *n*-hexane extract contained one major peak (Figure 1D); therefore, CPC was carried by calculation of its *K* value. Many peaks were revealed in the ethyl acetate extract, which was then purified on silica gel column chromatography. Detailed isolation methods were provided in Appendix A). Several repeated chromatographic methods, including CPC, open column chromatography, and prep-HPLC, led to a new diarylheptanoid (**4**) and four known sesquiterpenes (**1**, **2**, **5,** and **6**) and one diarylheptanoid (**3**) from *n*-hexane extract. In the EtOAc extract, six sesquiterpenes (**7**–**12**) were purified. The chemical structures of isolated compounds were elucidated as nootkatone (**1**) [18], eudsma-3,11-dien-2-one (**2**) [19], yakuchinone A (**3**) [20], 5′-hydroxy-yakuchinone A (**4**), alpinenone (**5**) [21], 6α-hydroxy-7-*epi*-cyperone (**6**) [22], (4*S**,5*E*,10*R**)-7-oxo-tri-*nor*-eudesm-5-en-4β-ol (**7**) [23], teuhetenone A (**8**) [24], 7-*epi*-teucrenone B (**9**) [24], 11-hydroxyvalenc-1(10)-en-2-one (**10**) [25], oxyphyllenodiol A (**11**), [26] and oxyphyllenodiol B (**12**) [26] based on spectroscopic data compared to the previously reported data.

Compound **4** was obtained as a yellow gum. Its molecular formula was determined as C_20_H_24_O_4_ based on the ESI-MS ion peak at *m/z* 327.4 [M-H]^−^ and ^1^H-, ^13^C-NMR. ^1^H-NMR spectrum of **4** was similar to that of yakuchinone A (**3**) except for the disappearance of one aromatic proton. In the ^1^H-NMR spectrum of **4** measured in CDCl_3_, five aromatic proton signals at δ_H_ 7.27 (2H, m, H-3′′, 5′′), 7.16 (3H, m, (H-2′′, 4′′, 6′′) showed a mono-substituted aromatic ring. The *meta*-coupled proton signals at δ_H_ 6.71 (1H, d, J = 2.1 Hz, H-6′) and 6.70 (1H, d, J = 2.1 Hz, H-2′) indicated the presence of a *meta*-coupled aromatic system in **4**. In CDCl_3_, *meta*-coupled proton peaks were too close to identify. However, the ^1^H-NMR spectrum measured in CD_3_OD showed well-separated proton peaks of H-2′ and H-6′ at δ_H_ 6.65 (1H, d, J = 2.0, H-2′) and 6.78 (1H, d, J = 2.0, H-6′), respectively (Figure 2).

Six methylene signals at δ_H_ 2.85 (2H, t, J = 7.5, H-1), 2.7 (2H, t, J = 7.5, H-2), 2.60 (2H, t, J = 7.1, H-7), 2.41 (2H, J = 7.0, H-4), 1.61 (2H, m, H-6), and 1.59 (2H, m, H-5) indicated the presence of a saturated aliphatic chain in **4**. ^13^C-NMR spectrum of **4** was similar that of yakuchinone (**3**), and ^13^C-NMR spectrum showed 12 aromatic carbons, one ketone carbon, and seven methylene carbons. Further analysis of the 2D-NMR spectra of **4** helped to define its molecular structure (Appendix A). The connections of the saturated aliphatic chain of H-1 (δ_H_ 2.85)/H-2 (δ_H_ 2.71) and H-4 (δ_H_ 2.41)/H-5 (δ_H_ 1.59)/H-6 (δ_H_ 1.61)/H-7 (δ_H_ 2.60) were confirmed by analysis of the ^1^H-^1^H COSY spectrum. The HMBC correlations of H-1 (δ_H_ 2.85) with C-1′ (δC 133.7), C-2′ (δC 110.8), C-6′ (122.8), and C-3 (δ 210.4) suggested that the aromatic rings were linked via a saturated aliphatic chain at C-1. In addition, the HMBC correlations of H-7 (δ_H_ 2.60) with C-1′′ (δC 141.0) and C-2′′/C-6′′ (δC 128.4) were also shown. The correlations of OCH_3_ (δ_H_ 3.90)/C-3′′ (δC 147.3.0), H-2′ (δ_H_ 2.71)/C-3′ (δC 147.3), C-4′ (δC 142.3), and H-6′ (δ_H_ 6.70)/C-4′(δC 142.3), C-5′ (δC 124.5) were proven to be substituted with a methoxy group in C-3′ and two hydroxy groups at C-4′ and C-5′ (Figure 3). From these results, the structure of compound **4** was assigned as a 5′-hydroxy-yakuchinone A.

Next, it was confirmed which compounds contributed to the increase in ARE-luciferase activity. The cytotoxic effects of compounds **1**–**12** were evaluated in HepG2 cells. No compounds showed any cytotoxicity when treated at concentrations of 3–30 μM for 24 h. As shown in Figure 4, compounds **2**, **3**, and **4** from the *n*-hexane extract enhanced the ARE-luciferase activities 22.6-fold, 2.9-fold, and 4.4-fold at 30 μM, respectively. Sulforaphane (5 μM) induced ARE enhanced activity 42-fold. Since compounds **7**, **9**, and **12** from EtOAc extract only slightly enhanced ARE-luciferase activity. Further studies were performed only using crude extract and compounds **2**–**4**.

### 3.2. Effects of Crude Extract and Active Compounds ***2**–**4*** against t-BHP-Induced Cell Death

The protective effects of crude extract and compounds **2**–**4** were tested against *t*-BHP-induced toxicity of HepG2 cells. *t*-BHP converted free radicals are often used to cause oxidative stress [6,7]. In this study, 300 μM of *t*-BHP showed cytotoxicity and reduced cell viability to ~20% of control levels, but pretreatment with crude extract or active compounds **2**–**4** for 12 h significantly prevented oxidative stress-induced cell death concentration-dependently (Figure 5). Crude extract increased cell viability to 63.4% at 30 μg/mL concentration. Its active compounds also protect HepG2 cells against *t*-BHP treatment. Compound **2** at 100 μM concentration increases cell viability from 26.3% to 79.9%. Compounds **3** and **4**, at a concentration of 30 μM, also improved the cell viability to 89.0% and 94.0%, respectively. In the induction of ARE-luciferase, compound **2** showed the most activity, but the protective ability of HepG2 cells against oxidative stress was better with compounds **3** and **4**. Therefore, free radical scavenging activities were compared by DPPH and ABTS assays.

### 3.3. Free Radical Scavenging Assay of Compounds (***2**–**4***)

ARE-inducing compounds **2**–**4** from *n*-hexane extract were evaluated by DPPH and ABTS radical scavenging activities at a concentration range of 3–100 μM. As shown in Figure 6, compound **3** exhibited the highest ABTS and DPPH radical scavenging activity, and compound **4** was next. However, compound **2** showed little DPPH and ABTS radical scavenging ability. Compared with induction of ARE-luciferase and free radical scavenging activities, compound **2** was considered to be an Nrf2 activator without radical scavenging ability, and compounds **3** and **4** behaved as free radical scavengers rather than ARE induction.

### 3.4. Nuclear Translocation of Nrf2 by Crude Extract and Compound ***2***

Compound **2** exhibited a protective effect against *t*-BHP-induced oxidative stress in HepG2 cells but did not scavenge free radicals in ABTS and DPPH assays. The compound **2** enhanced ARE-driven luciferase activity was described earlier. The nuclear translocation of Nrf2 was determined to confirm its ability to activate Nrf2. Treatment of compound **2** (30 μM) caused a 2-fold nuclear Nrf2 accumulation after 12 h exposure at a concentration of 30 μM (Figure 7A). The treatment of compound **2** at 30 μM increased nuclear Nrf2 levels 2.3-fold, and 5 μM sulforaphane enhanced levels 3.9-fold. Then 30 μM of compound **2** was treated, and as a result of checking the change in the level of Nrf2 protein expression with time, it was confirmed that the increase was the highest at 3 h after increasing from 1 h, and it was maintained until 12 h. Furthermore, the representative Nrf2-controlled antioxidant protein, HO-1, was also increased by compound **2** and crude extract (Figure 7C,D).

### 3.5. Optimization of Ethanol-Water Ratio for Extraction

Finally, the ethanol-water composition for extraction was optimized. The mixtures of ethanol and water were selected since these two solvents are preferable solvents due to safety considerations with health and handling.

We investigated the ethanol composition on extract characteristics such as radical scavenging (DPPH and ABTS) activity, ARE-induction effect, and HepG2 cell protective activity against *t*-BHP toxicity. In DPPH assays, 75% ethanol extract was the best, followed by 100% ethanol extract and 50% ethanol extract (Figure 8A). In the ABTS assay, 75% ethanol extract showed the highest efficacy. Both the 25% ethanol extract and 100% ethanol extract also exhibited radical scavenging activity, but the efficacy was different depending on the concentration (Figure 8B). When ARE-luciferase induction was tested, it was effective in the following order: 100% ethanol, 75% ethanol, and 50% ethanol (Figure 8C).

As a result, the 75% ethanol extract most effectively prevented the toxicity of HepG2 cells induced by *t*-BHP (Figure 8D). Therefore, 75% ethanol was selected as an extraction solvent for *A. oxyphylla* fruit.

After comparing antioxidant activity, each solvent extract was analyzed by HPLC to assess the amount of isolated compounds **1**–**12**. HPLC analysis revealed that nootkatone (**1**) and yakuchinone A (**3**) were major constituents in *A. oxyphylla*. Amount of eudesma-3,11-dien-2-one (**2**), active molecule for ARE induction was 4.36 (mg/g extract) in 75% ethanol extract and 3.96 (mg/g extract) in 100% ethanol extract. Yakuchinone A (**3**) was extracted from 75% and 100% ethanol. However, in the remaining solvent conditions, it was below the analysis range. Since eudesma-3,11-dien-2-one (**2**) and yakuchinone A (**3**) were active molecules against *t*-BHP-induced toxicity in HepG2 cells, these amounts were carefully checked. These two compounds were extracted from 100% ethanol and 75% ethanol (Table 1). 5′-Hydroxy-yakuchinone A (**4**) was also active, but the amount in all extracts was under the detectable range. Therefore, HPLC analysis made clear that eudesma-3,11-dien-2-one (**2**) and yakuchinone (**3**) were active markers in *A. oxyphylla* extract.

*A.**oxyphylla* fruit has been reported to exhibit antioxidant [27,28] and hepatoprotective effects against CCl_4_-induced oxidative damage via the Nrf2 pathway [29]. In addition, supercritical fluid extract of *A.*
*oxyphylla* exhibited a strong ferric reducing power at the high dosage of 1000 µg/mL, but the extract showed a minor level of Fe^2+^ scavenging effect [30]. 5-Hydroxymethylfurfural [30], protocatechuic acid [31], diarylheptanoids including yakuchinone A [32], and flavonoids [33] were reported as antioxidant active substances. Yakuchinone A was reported to possess DPPH radical scavenging and linoleic acid peroxidation inhibitory activities [34]. Nootkatone, the principal ingredient, was also reported to activate Nrf2 in the luciferase assay using transfected HCT-116 cells and increased HO-1 protein levels as a marker for antioxidant activity [28].

In this study, we found that *A. oxyphylla* extract protected HepG2 cells against *t*-BHP-induced toxicity in a dose-dependent manner. Activity-guided isolation led to the identification of 12 compounds, including a new diarylheptanoid named 5′-hydroxy-yakuchinone A (**4**). Among them, compounds **1**–**4**, **7**, **9**, and **12** increased ARE-luciferase activity at a concentration of 30 μM. Eudsma-3,11-dien-2-one (**2**) exhibited the highest effect on ARE induction. Yakuchinone A (**3**) showed the highest activities in DPPH and ABTS radical scavenging test. However, eudsma-3,11-dien-2-one (**2**) did not scavenge DPPH and ABTS radicals. Considering the strategies of protection against oxidative damage [3], eudsma-3,11-dien-2-one (**2**) increases endogenous antioxidant/phase II detoxifying enzymes via the Nrf2/ARE signaling pathway, and yakuchinone A (**3**) scavenge ROS as an antioxidant. The proposed mechanism of active compounds **2** and **3** was summarized in Figure 9.

## 4. Conclusions

*A. oxyphylla* extract significantly protects HepG2 cells against *t*-BHP-induced toxicity in a concentration-dependent manner. Bioassay-guided isolation led to the identification of two main active components, eudsma-3,11-dien-2-one (**2**) and yakuchinone A (**3**). Edesma-3,11-dien-2-one (**2**) promoted the nuclear accumulation of Nrf2 and increased the HO-l expression, and yakuchinone A (**3**) scavenged ABTS and DPPH radicals. Extraction solvent with mixtures of ethanol and water was also tested. Considering HepG2 cells protection, the ability of radical scavenging, ARE induction activities, and contents of active compounds **2**, **3**, 75% ethanol was chosen as the best extraction solvent. These results demonstrated that *A. oxyphylla* exerted antioxidant effects via the Nrf2/ARE pathway and radical scavenging, and the active markers were eudesma-3,11-dien-2-one (**2**) and yakuchinone A (**3**). Therefore, *A. oxyphylla* has the potential for use as an effective natural antioxidant for the prevention and treatment of oxidative stress-related liver diseases.

## Figures and Tables

**Figure 1 antioxidants-11-01032-f001:**
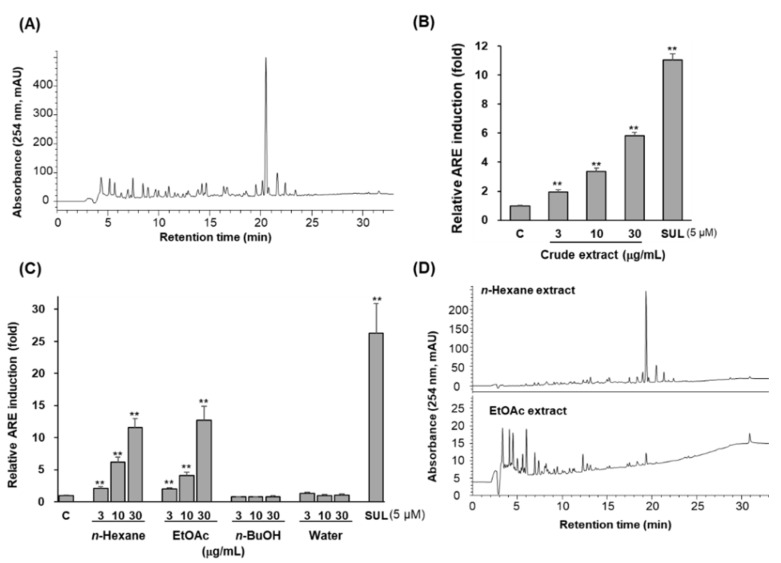
Crude extract of *A. oxyphylla* induced ARE-luciferase activity in ARE transfected-HepG2 cells. (**A**) HPLC chromatogram of crude extract of *A. oxyphylla*. (**B**) Relative ARE induction activity of crude extract (μg/mL). (**C**) Relative ARE activity of solvent partitioned extracts from crude extract. (**D**) HPLC chromatograms of *n*-hexane extract and EtOAc extract. Data are presented as the mean ± S.D. (*n* = 3). ** *p* < 0.01 (compared with the vehicle-treated control). SUL: sulforaphane was treated as a positive control.

**Figure 2 antioxidants-11-01032-f002:**
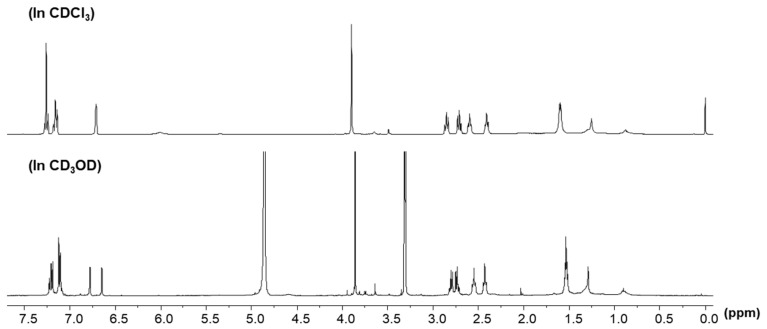
^1^H-NMR spectra of compound **4** in CDCl_3_ and CD_3_OD (400 MHz).

**Figure 3 antioxidants-11-01032-f003:**
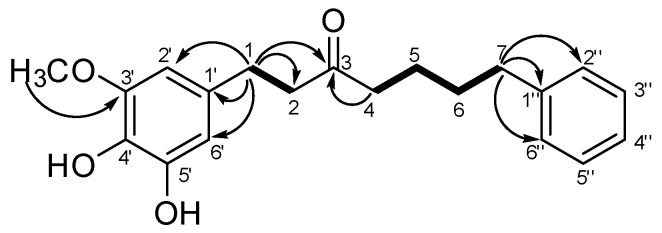
Key ^1^H-^1^H COSY (bold) and HMBC (arrows) correlations for compound **4**.

**Figure 4 antioxidants-11-01032-f004:**
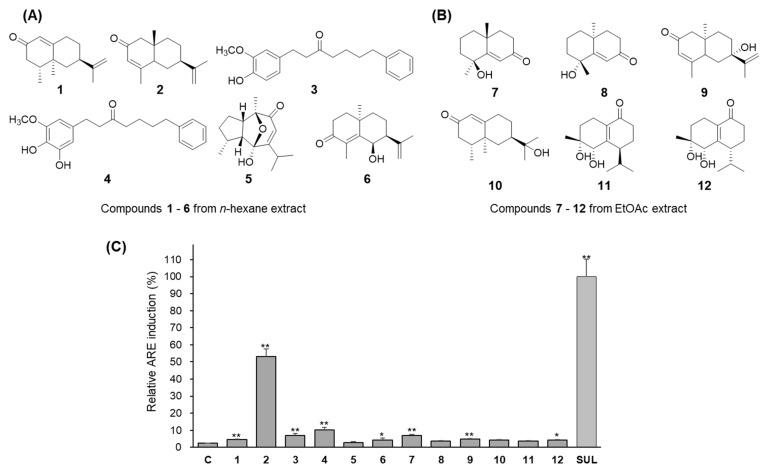
Chemical structures of isolated compounds **1**–**12** and their ARE-luciferase induction activity in HepG2 cells. (**A**) Compounds **1**–**6** were isolated from *n*-hexane extract. (**B**) Compounds **7**–**12** were purified from EtOAc extract. (**C**) The relative ARE induction activity of each compound to 5 μM sulforaphane (%). Each compound was treated at a concentration of 30 μM. Data are presented as the mean ± S.D. (*n* = 3). * *p* < 0.05, ** *p* < 0.01 (compared with the vehicle-treated control). Sulforaphane (SUL) was treated as a positive control.

**Figure 5 antioxidants-11-01032-f005:**
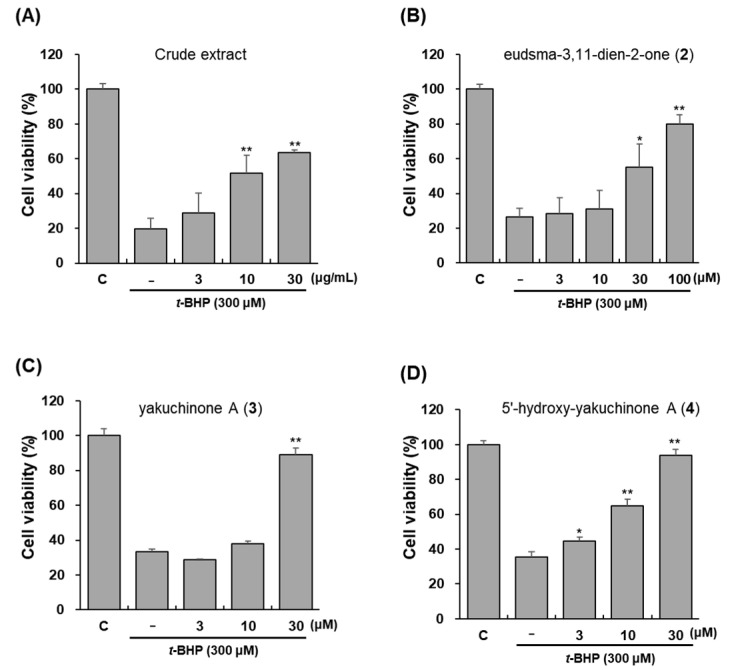
Protection against *t*-BHP-induced toxicity of total extract (**A**) and pure compounds **2**–**4** (**B**–**D**) from *A*. *oxyphylla*. HepG2-ARE cells were pre-treated with each compound (3, 10, 30, or 100 µM) for 12 h and added *t*-BHP (300 µM) for 4 h. Data are presented as the mean ± S.D. (*n* = 3). * *p* < 0.05, ** *p* < 0.01 (compared with the vehicle-treated control).

**Figure 6 antioxidants-11-01032-f006:**
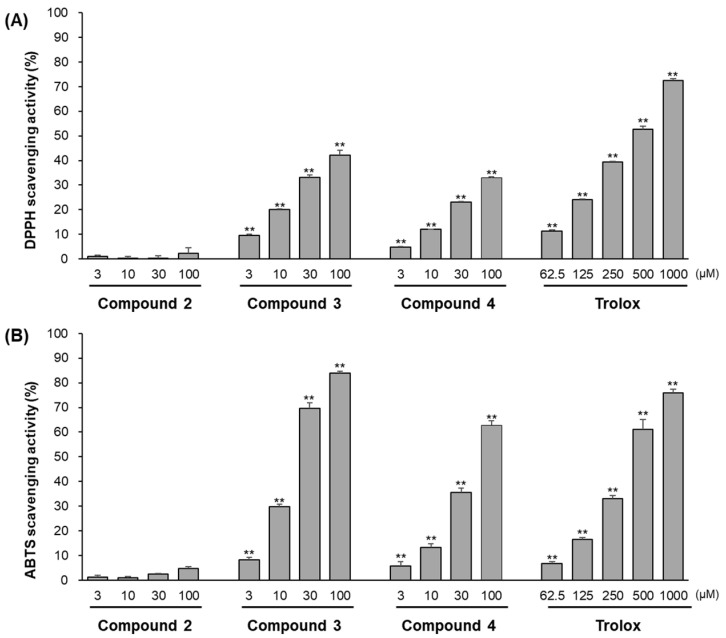
DPPH (**A**) and ABTS (**B**) radical scavenging activity of compounds **2**–**4**. The treated concentrations of compounds **2**–**4** were 3, 10, 30, and 100 μM. Trolox was used as a positive control. Data are presented as the mean ± S.D. (*n* = 3). ** *p* < 0.01 (compared with the vehicle-treated control).

**Figure 7 antioxidants-11-01032-f007:**
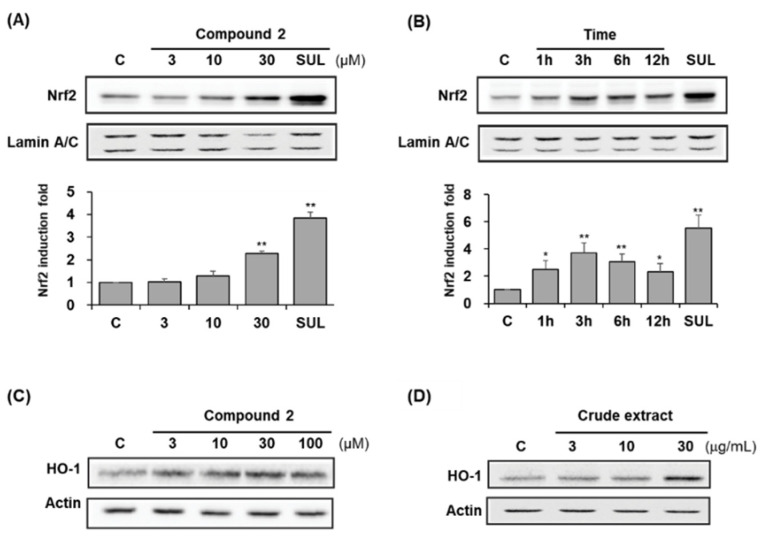
Induction of the nuclear Nrf2 protein and HO-1 levels in HepG2 cells by compound **2** or crude extract. Nuclear Nrf2 protein levels in HepG2 cells exposed to compound **2** for the indicated concentrations (**A**) and treated with 30 μM compound **2** for the indicated times (**B**). Western blot of HO-1 induced by treatment of compound **2** (**C**) and crude extract (**D**) for the indicated concentrations. Data are presented as the mean ± S.D. (*n* = 3). * *p* < 0.05, ** *p* < 0.01 (compared with the vehicle-treated control).

**Figure 8 antioxidants-11-01032-f008:**
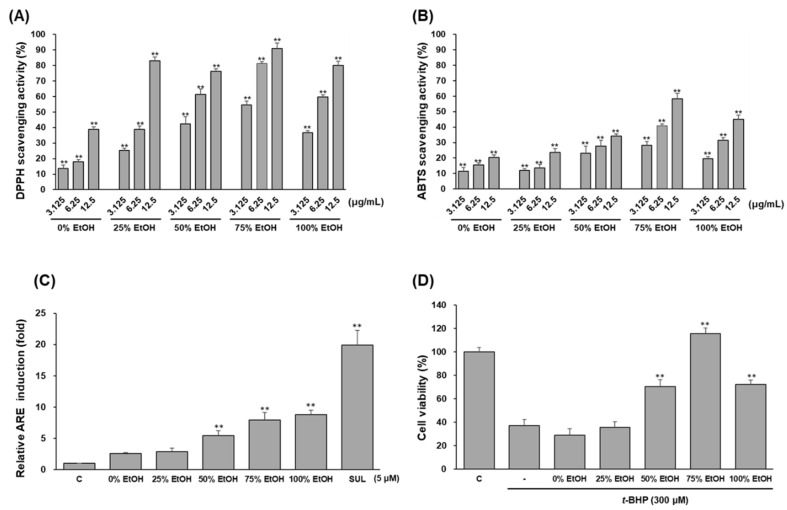
Effects of ethanol-water composition on the DPPH (**A**), ABTS (**B**) radical scavenging, ARE-luciferase induction (**C**), and HepG2 cell protective (**D**) activity. Data are presented as the mean ± S.D. (*n* = 3). ** *p* < 0.01 (compared with the vehicle-treated control).

**Figure 9 antioxidants-11-01032-f009:**
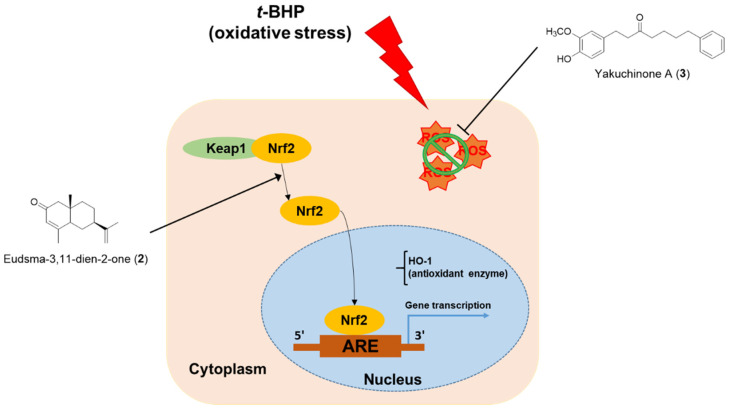
Proposed mechanism of active ingredients of *A. oxyphylla* for the protective activity against *t*-BHP-induced toxicity in HepG2 cells.

**Table 1 antioxidants-11-01032-t001:** Contents of compounds **1**–**12** in extract obtained from mixtures of ethanol and water.

Extraction Solvents	100% Ethanol	75% Ethanol	50% Ethanol	25% Ethanol	0% Ethanol
Extraction yield (g/100 g of sample)	6.53 ± 0.41	9.51 ± 0.64	10.88 ± 1.14	9.81 ± 0.94	11.57 ± 0.26
Compounds	Content (mg/g extract)				
Nootkatone (**1**)	22.07 ± 0.149	24.78 ± 0.117	1.48 ± 0.003	1.06 ± 0.006	0.79 ± 0.005
Eudesma-3,11-dien-2-one (**2**)	3.96 ± 0.032	4.36 ± 0.026	0.06 ± 0.001	0.03 ± 0.001	n.d.
Yakuchinone A (**3**)	9.60 ± 0.109	9.06 ± 0.094	n.d.	n.d.	n.d.
5′-Hydroxyl-yakuchinone A (**4**)	n.d.	n.d.	n.d.	n.d.	n.d.
Alpinenone (**5**)	1.63 ± 0.008	1.81 ± 0.005	1.06 ± 0.015	0.89 ± 0.004	0.70 ± 0.001
6α-Hydroxy-cyperone (**6**)	0.99 ± 0.007	1.12 ± 0.004	0.30 ± 0.007	0.20 ± 0.004	0.11 ± 0.001
(4*S*, *5E*, 10*R*)-7-Oxo-tri-nor-eudesm-5-en-4β-ol (**7**)	3.02 ± 0.014	3.22 ± 0.002	1.88 ± 0.015	1.78 ± 0.012	1.19 ± 0.002
Teuhetenone A (**8**)	2.58 ± 0.015	2.72 ± 0.017	1.75 ± 0.023	1.55 ± 0.010	1.13 ± 0.004
7-*epi*-Teucrenone B (**9**)	n.d.	n.d.	n.d.	n.d.	n.d.
11-Hydroxyvalenc-1(10)-en-2-one (**10**)	3.14 ± 0.015	3.47 ± 0.004	1.74 ± 0.016	1.56 ± 0.008	1.20 ± 0.002
Oxyphyllenodiol A (**11**)	2.64 ± 0.007	2.63 ± 0.014	2.41 ± 0.015	2.32 ± 0.009	2.03 ± 0.004
Oxyphyllenodiol B (**12**)	4.73 ± 0.031	5.04 ± 0.035	2.95 ± 0.049	2.52 ± 0.022	1.62 ± 0.009

n.d.: not detectable.

## Data Availability

Data is contained within the article or Appendix A.

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
