# Peer review of "Protective Effect of Alpinia oxyphylla Fruit against tert-Butyl Hydroperoxide-Induced Toxicity in HepG2 Cells via Nrf2 Activation and Free Radical Scavenging and Its Active Molecules"

_antioxidants, 2022, doi:10.3390/antiox11051032_

Round 1

Reviewer 1 Report

This is a serious and classical study of antioxidant properties of some compounds extracted from a Chinese fruit. Many natural compounds have been studied this way. Thus it can be published.

Some compounds belong to the well-known families of polyphenols (e.g. Compound 4). Their beneficial properties are known. They usually react with free radicals such as DPPH quite fast. However, the most interesting properties are not the scavenging of DPPH or other models, their action on Nrf2 is much more interesting. As for free radicals, It would be much more interesting to study their reaction with superoxide (for instance using the xanthine/xanthine oxidase system). I suggest the authors add it in their paper if possible.

There is no study of the harmful effects of these compounds. An easy one would be the scavenging of iron. I suggest the authors make somme attempts in this direction. 

Minor:`

There are numerous non classical abbreviations. A list of abbreviations should be added.

The details of NMR should all be in supplementary material
p. 10 why a paragraph in bold characters?

Author Response

This is a serious and classical study of antioxidant properties of some compounds extracted from a Chinese fruit. Many natural compounds have been studied this way. Thus it can be published.

Some compounds belong to the well-known families of polyphenols (e.g. Compound 4). Their beneficial properties are known. They usually react with free radicals such as DPPH quite fast. However, the most interesting properties are not the scavenging of DPPH or other models, their action on Nrf2 is much more interesting. As for free radicals, It would be much more interesting to study their reaction with superoxide (for instance using the xanthine/xanthine oxidase system). I suggest the authors add it in their paper if possible.

There is no study of the harmful effects of these compounds. An easy one would be the scavenging of iron. I suggest the authors make some attempts in this direction. 

- Previous studies revealed that A. oxyphylla extract exhibited a strong ferric reducing power at the high dosage of 1000 μg/ml (0.52 ± 0.01) in contrast to low concentration at 50 μg/mL (0.16 ± 0.01), and the positive control was BHA at 100 μM. In addition, the extract presented minor level of Fe2+ scavenging effect.

- Yakuchinone A exhibited DPPH radical scavenging effect and linoleic acid peroxidation inhibitory activity

We add these previous reported data in discussion section and add references 30 and 35.

Minor:

There are numerous non classical abbreviations. A list of abbreviations should be added. The details of NMR should all be in supplementary material

- We add full name for abbreviation in Abstract and Main text and marked in blue color. And NMR and Mass spectral data moved in Supplementary Materials.

  1. 10 why a paragraph in bold characters?

- Corrected.

Reviewer 2 Report

The manuscript submitted by Dr. XXX and co-workers focuses on the antioxidant effect of Alpinia oxyphylla fruit extract. The authors characterize the plant extract and study the antioxidant effects, as well as other biological properties in a different way. Several bioactive compounds have been identified and the way in which they protect the cell lines under study from the oxidative stress generated by model organic peroxide is highlighted.

The introduction makes an appropriate transition to the studies undertaken. The results are numerous and support the conclusions presented from several points of view.

From my point of view it is a complete study and can be published in this form. Minor changes are possible that do not affect the shape and content of the material. For example, the authors should specify in the material what other studies have been done on the antioxidant capacity of Alpinia oxyphylla extracts and somewhat correlate the results.

Author Response

The manuscript submitted by Dr. XXX and co-workers focuses on the antioxidant effect of Alpinia oxyphylla fruit extract. The authors characterize the plant extract and study the antioxidant effects, as well as other biological properties in a different way. Several bioactive compounds have been identified and the way in which they protect the cell lines under study from the oxidative stress generated by model organic peroxide is highlighted. The introduction makes an appropriate transition to the studies undertaken. The results are numerous and support the conclusions presented from several points of view.

From my point of view, it is a complete study and can be published in this form. Minor changes are possible that do not affect the shape and content of the material. For example, the authors should specify in the material what other studies have been done on the antioxidant capacity of Alpinia oxyphylla extracts and somewhat correlate the results.

- We add these previous reported data in discussion section and add references 30 and 35. And our research was foscued that antioxidants effects of Alpinia oxyphyll were via Nrf2/ARE pathway and radical scavenging, and the active markers were eudesma-3,11-dien-2-one (2) and yakuchinone A (3).